# Fabrication and Compressive Behavior of a Micro-Lattice Composite by High Resolution DLP Stereolithography

**DOI:** 10.3390/polym13050785

**Published:** 2021-03-04

**Authors:** Chow Shing Shin, Yu Chia Chang

**Affiliations:** Department of Mechanical Engineering, National Taiwan University, No. 1, Sec. 4, Roosevelt Road, Taipei 10617, Taiwan; R06522531@ntu.edu.tw

**Keywords:** 3D printing, micro-lattice material, photo-polymerization, DLP micro-stereolithography, electroless nickel plating, compressive behavior

## Abstract

Lattice structures are superior to stochastic foams in mechanical properties and are finding increasing applications. Their properties can be tailored in a wide range through adjusting the design and dimensions of the unit cell, changing the constituent materials as well as forming into hierarchical structures. In order to achieve more levels of hierarchy, the dimensions of the fundamental lattice have to be small enough. Although lattice size of several microns can be fabricated using the two-photon polymerization technique, sophisticated and costly equipment is required. To balance cost and performance, a low-cost high resolution micro-stereolithographic system has been developed in this work based on a commercial digital light processing (DLP) projector. Unit cell lengths as small as 100 μm have been successfully fabricated. Decreasing the unit cell size from 150 to 100 μm increased the compressive stiffness by 26%. Different pretreatments to facilitate the electroless plating of nickel on the lattice structure have been attempted. A pretreatment of dip coating in a graphene suspension is the most successful and increased the strength and stiffness by 5.3 and 3.6 times, respectively. Even a very light and incomplete nickel plating in the interior has increase the structural stiffness and strength by more than twofold.

## 1. Introduction

Enabling micro-distribution of material in a solid to form a cellular material is a useful approach to extend the range of usable mechanical properties of engineering materials. In the past decades, cellular materials have found increasing applications in impact energy absorption [1,2,3,4], vibration damping [5,6], fluid filtering [3,7], heat exchanging [3,7], biomedical implants [7,8,9] and scaffolds [10,11,12] as well as in structural components [13,14]. A widely known cellular material form is foam whose stochastic structure cannot guarantee optimum distribution of material at the positions most needed to enhance a certain property. Moreover, its deformation is mostly dominated by the bending of the cell walls and flexural stiffness is often inferior to that under axial loading. It has been shown that lattice materials with regularly repeating truss-like unit cells possess superior mechanical properties to foams [15,16,17,18,19,20]. Their properties can be tailored by adjusting not only the type and size of trusses constituting the unit cells but also the dimensions of individual struts in the trusses. As a result, the mechanical properties of lattice materials are much more flexible to customization and optimization than foams. A number of different trusses such as tetrahedral [20,21,22,23], pyramidal [24], Kagome [23,25,26], octet [27,28,29], body-centered cubic and face-centered cubic [30] have been studied. Early experimental investigations employed casting [25], deformation forming of perforated metal sheet [15,31,32], metal wire weaving [33,34] and fitting together of precut truss elements [35] to produce the lattice structures. Their fabrication by these traditional manufacturing techniques are complicated, costly and the achievable lattice designs are limited. Some of these techniques are wasteful of raw materials and some involve severe deformation that makes them suitable only for very ductile materials. Moreover, traditional manufacturing techniques typically give large lattice cell sizes in the centimeter range. With the advances in additive manufacturing (AM), lattice with complex geometry can be very quickly and efficiently fabricated. With some high-resolution techniques, the cell size can go down to the micron level. Smaller cell sizes allow hierarchical lattice structures to be more readily fabricated [36,37].

Each of the seven categories of AM processes, as defined in the ISO/ASTM 52900:2015 standard, have been employed to fabricate lattice materials [38,39]. Of these, the vat polymerization technique has the following advantages over other processes: (1) unlike the powder-based methods, there is no trapped powder sticking to the inner surface of the printed structure. Any excess material is in the form of uncured liquid resin which is relatively easy to remove; (2) temporary support structures may sometimes be required in the course of additive fabrication of some lattice designs. When trapped inside the lattice, these support structures are difficult to remove. In vat polymerization, such structures are often not necessary as support can be provided through flotation by the liquid resin; (3) the properties of the resulting printed structure can easily be modulated by adding reinforcing agents or other ingredients into the resin to produce a composite material [40]. For example, the material may be made ferromagnetically or electrically conductive by adding magnetic [41,42,43] or conductive [40,44,45,46] particles to the resin; (4) further properties modulation can be achieved by using the printed polymer structure as a template to produce lattice made of metallized polymeric composites [47], metal or ceramic hollow tubes or solid ceramic lattice [48,49,50]. Metallization of the surface of a polymeric lattice increased its stiffness and strength [47].

Commonly employed vat polymerization processes include two-photon polymerization and single photon stereolithography. The former involves the simultaneous absorption of two photons to excite a photo-initiator atom into a radical, thereby initiating polymerization. Light from a femtosecond laser is usually focused into a fine spot to get the extremely high intensity required to bring about this non-linear optical process [51]. Polymerization can only occur locally inside the focus voxel and so a sub-micron resolution can be readily achieved [49,50]. This technique requires expensive advanced equipment. The point-by-point scanning of sub-micron voxel limits both the speed and size of the fabrication. When high energy ultraviolet or near ultraviolet photons are employed, polymerization can be initiated by single photon absorption. Point-by-point scanning with a laser beam [40,52] or projection with a digital mirror device (DMD) [48,53] have both been used for this manufacturing process. A DMD consists of an array of millions of independently switchable micro-mirrors to form an image. Polymerization of a whole layer occurs in one projection and so it can fabricate much faster than point-by-point scanning. However, as light reflected from a micro-mirror has an intensity distribution spreading beyond the intended pixel, when a number of bright pixels cluster together, the superposition of light energy in the adjacent dark pixels may be high enough to bring about polymerization. As the lattice unit cell becomes very small, the strut elements will be crowded together. Empty spaces between the struts may become filled by the above unintended “dark curing”. The smallest micro-lattice cell sizes produced by this technique are reported as typically being several hundred microns in length [48,53].

With smaller lattice cell size, more cells and structural elements can be packed into the same volume. Moreover, smaller cell size leads to a higher surface to volume ratio and more effective surface metallization. This should result in more marked strengthening effects. To test the above hypothesis, a low-cost high-resolution micro-stereolithographic system based on a commercial digital light processing (DLP) projector will be developed. Micro-lattice with cell size as small as a hundred microns will be attempted to investigate the effect on stiffness and failure strength of cell size as well as surface metallization with electroless nickel plating.

## 2. Materials and Methods

### 2.1. Micro-Stereolithographic Printing System

A commercial DLP projector (P1500, ACER, Taipei, ROC) is employed to project onto a moving platform for printing. The platform was driven by a stepping motor operated ball screw (Figure 1) and travels vertically with a resolution of 0.5 μm. This projector is equipped with the DarkChip 3 digital mirror device (DMD) which has a resolution of 1920 × 1080 pixels. To reduce the size of the projected image, the lens of the projector was inverted and positioned so as to project the image through a 10× microscope objective with a numerical aperture of 0.25 (LMU-10X-NUV, Thorlabs, NJ, USA). The resulting image from the objective measured 1.69 × 0.95 mm^2^, giving a resolution of 0.88 μm/pixel on the printing platform. The image was focused on the surface of the liquid resin bath. The platform was first located just beneath the liquid surface. After each exposure a layer of resin was cured with the pattern projected. The platform was then lowered into the resin bath by a definite amount for the next exposure. The process was repeated to build up a three-dimensional structure consisting of a stack of patterned layers that constitute the designated micro-lattice structure. After printing, uncured resin is washed off the structure using ethanol.

### 2.2. Lattice Design

Deshpande et al. compared a number of different truss configurations and concluded that the octet-truss has more favorable strength and stiffness than others as it has tensile members to resist deformation under different loading conditions [27]. A truss structure (Figure 2) similar to that of the octet-truss was employed as the repeating lattice unit in the current work to produce 0.8 mm × 0.8 mm × 1.5 mm prints. A three-dimensional structure with this repeating unit was first drawn up and was then sliced into successive layers. The layer patterns were projected sequentially on to the resin to form the structure.

### 2.3. Photo-Curable Resin

A resin consisting of aliphatic acrylate monomers and acrylate oligomers with phosphine oxide-based photo-initiator was employed. The resin was optimized to cure using the DLP projector light and was supplied under the brand name “Deep Black” by FunToDo Resins (Alkmaar, The Netherlands). The manufacturer reported that the resin is sensitive to light with wavelengths from 225 to 415 nm.

The current additive fabrication cures the resin layer by layer. With a coarse layer thickness, slanting members in the lattice truss structure will be difficult to form. Printing with a small layer thickness is highly time consuming. Moreover, the penetration of projected light into the resin may lead to curing at depths below the intended surface curing layer, resulting in unwanted material in the empty space of the lattice. A compromise in the layer thickness is needed to alleviate the above problems. To limit the curing depth, one can control the exposure time as well as adding suitable light blockers to the resin. A number of ultraviolet and blue-light blockers have been tested and it was found that Sudan I gave the most satisfactory result. Moreover, trial and error with a series of exposure times, Sudan concentrations and layer thicknesses showed that successful printing of the truss structure can be achieved for exposure times of 600–900 ms, Sudan I concentrations of 0.35–0.45 g/100 mL resin and layer thickness of 3–5 μm. Since the projected light intensity tends to fade away from the center of the image, the corners of the image would fail to print with 0.45 g/100 mL Sudan concentration and a 5 μm layer thickness. An optimum parameter combination chosen for printing the 150 and 130 μm cells is 0.4 g Sudan I per 100 mL resin, a 4 μm layer thickness and exposure time of 650 ms. For the 100 µm cell, the Sudan I concentration is increased to 0.45 g/100 mL to ensure successful printing.

### 2.4. Nickel Plating of the Micro-Lattice Structure

Since electroless nickel will not plate directly onto the polymer surface, pretreatment to coat a thin conductive layer is needed. Four different pretreatment methods have been compared: (1) dip coating under sonication in an aqueous suspension with dispersed 0.3 wt% of carbon nanotube (CNT) for 3 min; (2) dip coating under sonication in a aqueous suspension with dispersed 0.3 wt% of graphene for 3 min; (3) soaking in an activating palladium chloride solution for 1 day; (4) coating a thin layer of aluminum by sputtering. Method 3 is the conventional pretreatment for plating on plastic components. It is fairly time consuming and so the more expedient Methods 1 and 2 are attempted with easily available conducting nano materials. Sputtering is known to reliably coat a very thin layer of metal on a component surface and so it is also employed in Method 4 for comparison. After pretreatment, excessive material is rinsed off and the structure is dried before immersing into a commercial electroless nickel plating formula containing nickel sulfate and hypophosphite (EN-430, Sendanex, Taipei, ROC). To ensure good penetration of the plating solution into the structure, the mixture is sonicated using an ultrasonic bath for 3 min. It is then continuously stirred and maintained at 90 °C on a magnetic stirrer during the whole plating process. The space between truss elements in the lattice structure with unit cell length of 100 µm is too small for effective plating to take place and the 130 µm lattice is employed for plating. Since the effects of various pretreatments are different, the structures from different pretreatments are soaked in the plating solution for different lengths of times in order to achieve a plating thickness of 3–5 μm.

### 2.5. Compression Testing of the Micro-Lattice Structure

The structures were compression tested between the printing platform and a low force load cell (FS2050-0000-1500-G, TE Connectivity, Schaffhausen, Switzerland). Displacement of the platform was monitored with a linear variable differential transformer (DFg1, Solartron Metrology, West Sussex, UK). The load-displacement trace was recorded.

## 3. Results and Discussion

### 3.1. As printed Structure with Different Lattice Sizes

Figure 3 shows the top views of the micro-latticed structures printed with unit cell lengths 150, 130 and 100 µm. Table 1 lists the dimensions of the unit cell lengths and widths of the key truss elements of these lattices measured from the magnified views from a scanning electron microscope (SEM).

The measured dimensions are all larger than those designed. One reason for this is the fine adjustment in the focal plane for printing results in a projected image slightly larger than that in the design drawing. Another reason is the occurrence of “dark curing” mentioned before. This is further illustrated schematically in Figure 4. Suppose the center white square in Figure 4 is the designed cross section of a rod to be formed by stacking a number of layers from the bottom upward. The width of the rod constitutes a number of pixels. Although each pixel of the image corresponds to a micro-mirror on the DMD that can independently be turned on or off, the light reflected from each micro mirror is a Gaussian distribution with intensity spreading well beyond the pixel [53]. When a number of bright pixels cluster together, the resulting superposed light intensity distribution may look like the dotted line in Figure 4. The superposed intensity at some dark pixels adjacent to the edge of the rod may be high enough to cure the resin there (such as D1 in Figure 4), leading to part of the observed increase in width shown in Table 1. Although the depth of curing is intended to be the chosen printing layer thickness, light may still penetrate to greater depths with decreasing intensity. Repeated illumination with the weak intensity light responsible for dark curing may accumulate enough energy to bring about polymerization at depths below the current printing layer. In Figure 4, D2–D4 represent layers with progressively more accumulation of this energy and so the extent of curing is bigger in deeper layers. The latter effect may be the reason that the element with square cross-section (the one with width A3 in Figure 2) appears round in the SEM micrographs (e.g., Figure 3c).

Past experience shows that if the UV blocker is not doing an adequate job under the chosen layer thickness and exposure time, the resulting structure may have an uppermost layer with well-defined truss elements such as those in Figure 3, while the interior, which suffers a higher number of repeated illuminations, may have its empty space partly or fully filled up. Sectioning opens the structure at about mid-height confirming that intended truss elements and empty spaces prevail through the height of the structure (Figure 5).

### 3.2. Electroless Nickel Plating of the Micro-Latticed Structure

Figure 6 shows the effect of electroless nickel plating following four different pretreatments. Table 2 compares the measured truss element widths before and after plating while Table 3 lists the atomic % of element detected from energy dispersive spectroscopy (EDS) analysis on the plated structure. Pretreating with CNT seems to enable plating to occur extensively (Figure 6a). However, the truss element width has only increased ~3 μm after 1 h of plating (Table 2). An EDS analysis indicates the occurrence of ~17.5 atomic % of nickel on the uppermost layer of the micro-lattice structure on the region shown in Figure 6a. This suggests the plating has incomplete coverage of the truss element even on the exterior layer. Table 2 shows the thickness of the nickel layer is about one-sixth of the polymeric truss, thus the amount of nickel on a cut open section is expected to be one-sixth that on the exterior layer. EDS analysis of the interior only yields ~1.6 atomic % of nickel. This suggests that the CNT or nickel-plating solution has difficulty getting into the interior. The excessive material deposited seen in Figure 6a may be attributed to the high aspect ratio of the CNTs that easily entangle with each other and straddle across two closely spaced truss elements near a joint, inducing nickel plating there.

Pretreating with graphene (Figure 6b) and sputtering with aluminum (Figure 6c) give a much smoother plating surface. Plating is also faster with these treatments. Table 2 shows that the truss widths have increased by ~5–8 μm while the plating duration is 1 h with graphene coating and 0.5 h with the aluminum sputtering. The atomic % of nickel on the exterior layers are respectively, ~53% and ~43%, which are significantly higher than that with CNT coating. The atomic % of nickel in the interior is again disproportionately low, being 3.4% and 0.56%, respectively, for graphene and aluminum coated cases. The exceptionally low level of nickel in the latter case may probably be due to the fact that the truss structure presented many obstacles to prevent the sputtered aluminum atoms to get into the interior. The sparingly coated interior can therefore hardly get plated.

Nickel plating after pretreating with palladium chloride (Figure 6d) is the slowest of the four and it took 3.5 h to add ~3–4 μm. Nickel content is ~40 atomic % on the exterior layer and ~3% in the interior. The plated surface has comparable smoothness as the graphene treated case.

In all four cases, nickel coverage on the outermost layers is not 100%. Pretreating with graphene and palladium chloride have better overall plating performance than CNT and aluminum sputtering. Plating has difficulty in getting into the interior in general. Judging from the interior nickel contents under different treatments, it may be inferred that the difficulty of the treatment materials to get inside has more responsibility than that of the plating solution for the under-plating of the interior.

### 3.3. Compressive Load-Displacement Behavior of the Different Micro-Lattice Structures

The compressive behavior under three repeated loading-unloadings of the micro-lattice structures with different unit cell sizes up to about ~20% strain is shown in Figure 7.

The initial stiffness is relatively low but it increases after a certain amount of deformation. When the lattice truss is compressed, bending of the slant members may accommodate the initial compressive deformation, leading to a low initial stiffness. Further compression will eventually start to stretch the horizontal members, giving rise to an increased stiffness. On fully unloading, the structures took a few seconds to return to zero deformation. This may be attributed to the viscoelastic behavior of the polymer. The fact that load-displacement traces followed the same hysteresis loops on repeated loading-unloading suggests the structure remained elastic without noticeable damage up to a deformation of ~20%. Figure 8 plots the load-displacement traces of the micro-lattice structures with different unit cell sizes together. Table 4 lists the stiffnesses computed after a strain of 10% of these structures. The stiffness steadily increases from 8.8 to 11.1 MPa as the unit cell size decreases from 150 to 100 μm. This may be attributed to the increased number of unit cells and truss elements to bear the load within the same volume as the cell size is reduced.

The nickel-plated structures all have higher stiffness than the unplated structure (Figure 9). Moreover, like the unplated structure, the load-displacement traces followed the same hysteresis loops on repeated loading-unloading, indicating that the structures remained elastic without noticeable damage up to a deformation of ~20%. Different pretreatment before plating led to different surface plating qualities and different amounts of nickel plating on to the interior of the structure as pointed out before. Table 5 compares the stiffnesses computed after a strain of 10% of these structures. The graphene treated structure exhibited the largest stiffness of 34.2 MPa, which is 3.6 times that of the unplated one. As metallic nickel has a considerably higher Young’s modulus and strength than the cured resin, more coverage of the nickel plating will help to improve the stiffness of the plated structures. The stiffness of the differently pretreated structures are in the order: graphene treated > palladium activated > CNT treated > aluminum sputtered. This order is the same as the amount of nickel content in the interior (Table 3), which also reflects the coverage of nickel plating.

### 3.4. Compressive Failure of the Different Micro-Lattice Structures

Figure 10 shows the compressive load-displacement traces to failure of the as-printed micro-lattice structures with different unit cell sizes. Following the initially lower stiffness for the first 0.1 mm compression, stiffness increased and maintained a roughly constant value until approaching the maximum load. The steep increase in stiffness before failure due to compacting, typically observed in cellular structures, did not occur in the current cases. In fact, monitoring through optical microscope shows that throughout the compression tests′ compacting of the lattice did not occur. The structures failed by buckle and bending, giving rise to the abrupt drop in loading following the maximum load. This mode of failure occurs probably because of the very slender aspect ratio of the compression specimens fabricated due to limitations of the current set up. If structures with more favorable aspect ratios are fabricated, it is expected the failure loads will become significantly higher as compaction of structures will have the chance to take place. Nevertheless, the current test results help to shed some light on the relative strengths of different treatments. The failure load of the 130 μm unit cell is slightly higher than that of the 150 μm while comparable to the 100 μm structure. Presumably, these are due to the same reasons that affect the order of the stiffness as explained before.

Typical fracture surfaces of the compressive failed micro-lattice structures (Figure 11) show clearly that there is no compacting and fracture at the joints of the truss occurred extensively.

Nickel plating has considerably increased the failure strength of the 130 μm structure as is evident in Figure 12. The corresponding strains at failure have also been extended to markedly larger displacements. Failure strengths and strains are listed in Table 6. In the best case of graphene pretreatment, failure strength has been increased by 5.3 times. In the worst case of aluminum sputtering, it has been increased by 2.4 times. Again, the order of the failure loads for different pretreatment is the same as the amount of nickel content in the interior for that treatment as listed in Table 3. This corroborates with the above explanation that nickel has a higher strength than resin and more plating coverage leads to higher strength in the metallized lattice structure. It is worth noting that from the relative ratio of the material, the interior of the structure probably contributes more to the exhibition of stiffness and strength. The current results indicate that even a very light and probably incomplete nickel plating in the interior already has a considerable beneficial effect on the structural stiffness and strength. A more thorough plating is expected to lead to much better improvement in mechanical properties. Song et al. have also fabricated stereolithographic lattice material composite metallized with nickel using the palladium chloride pretreatment [47]. Direct comparison of the stiffnesses and strengths with the current results may not be meaningful as the resin materials and the lattice designs are different. However, it is interesting to note that they reported a 1.29 and1.38 times increase in stiffness and strength, respectively, after nickel plating. These increases are considerably smaller than those observed in the current case. A possible reason is their large lattice unit cell (~7.5 mm in length) and struts (diameter ~1 mm) limited the surface area and led to a much smaller nickel to resin ratio, so nickel contributed much less towards increasing the stiffness and strength.

## 4. Conclusions

A low-cost micro-stereolithographic system based on a commercial DLP projector has been developed. It has been demonstrated that a micro-lattice based on a truss unit similar to the octet truss with cell length as small as 100 μm can successfully be fabricated with this system. By reducing the cell size from 150 to 100 μm increased the compressive stiffness from 8.8 to 11.1 MPa. Nickel plating of the resulting printed structure has markedly increased its stiffness and failure strength. Pretreating with graphene or activating with palladium chloride allows much better subsequent nickel plating than treating with CNT or sputtering with aluminum. Although the amount of nickel plated into the interior of the structure is sparse and incomplete, failure strength and stiffness has been increased by 5.3 and 3.6 times, respectively, in the best case of graphene pretreatment, and 2.4 and 2.3 times, respectively, in the worst case of aluminum sputtering. The results of the current work shows that it can be fabricated in a low-cost way to high resolution and that its mechanical properties are not limited by the cured resin but can be easily improved to a large extent by surface metallization. The small cell size also facilitated the fabrication of hierarchical lattice structures, which is a further way to modulate the mechanical properties. With the availability of a wider property range, their applications in impact energy absorption, vibration damping, biomedical implants and structural designs that require being light weight can become more flexible.

## Figures and Tables

**Figure 1 polymers-13-00785-f001:**
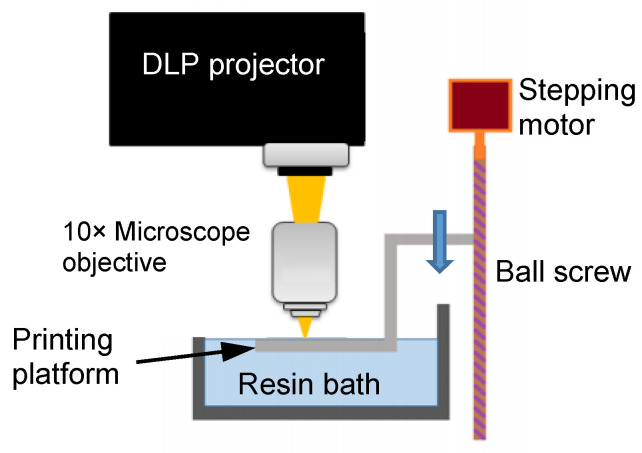
The digital light processing (DLP) printing system.

**Figure 2 polymers-13-00785-f002:**
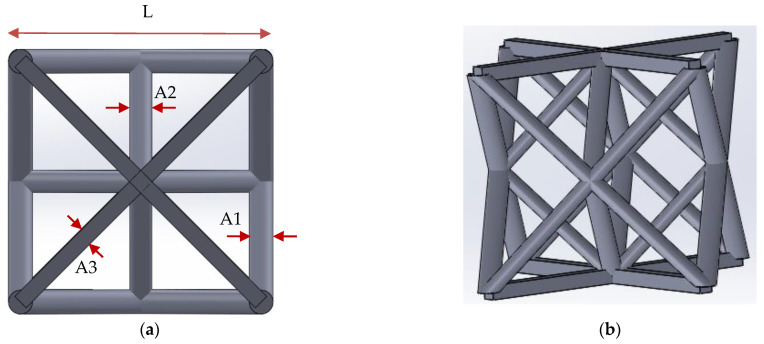
(**a**) Top view and (**b**) side view of the micro-lattice truss unit cell.

**Figure 3 polymers-13-00785-f003:**
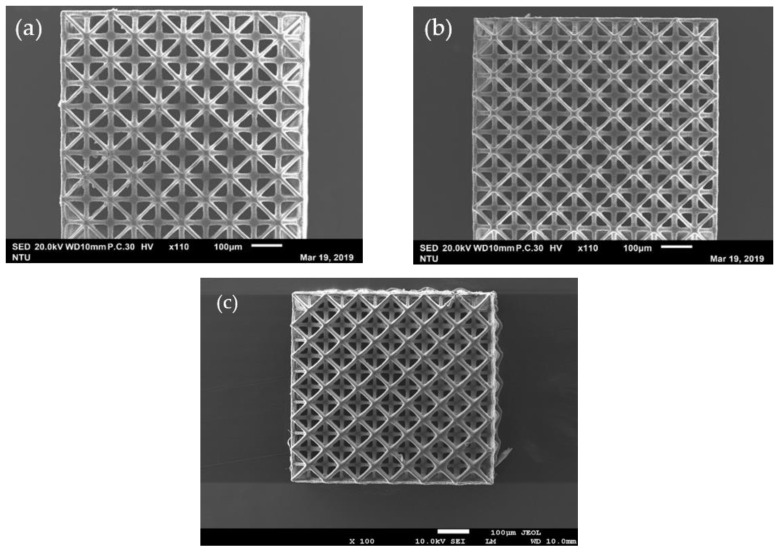
Top view of micro-lattice structures with unit cell length (L) of (**a**) 150 μm; (**b**) 130 μm; (**c**) 100 μm.

**Figure 4 polymers-13-00785-f004:**
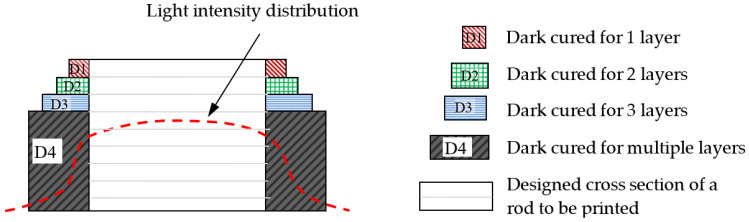
Schematic illustration on the effect of dark curing on the printed dimension of a square section.

**Figure 5 polymers-13-00785-f005:**
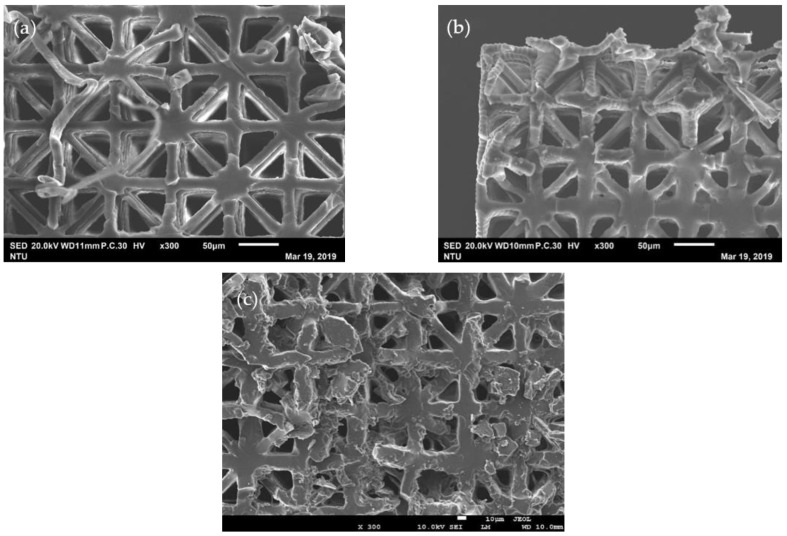
Interior view of the micro-lattice structures with unit cell length L of (**a**) 150 μm; (**b**) 130 μm; (**c**) 100 μm, cut open at about mid-height.

**Figure 6 polymers-13-00785-f006:**
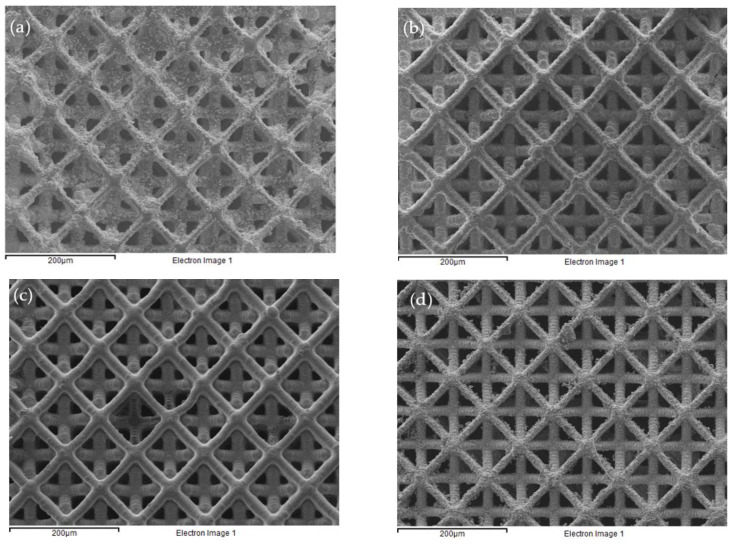
Top view of the electroless plated micro-lattice structures following pretreatment of (**a**) dip coating in carbon nanotube; (**b**) dip coating in graphene; (**c**) aluminum sputtering; (**d**) palladium chloride activating.

**Figure 7 polymers-13-00785-f007:**
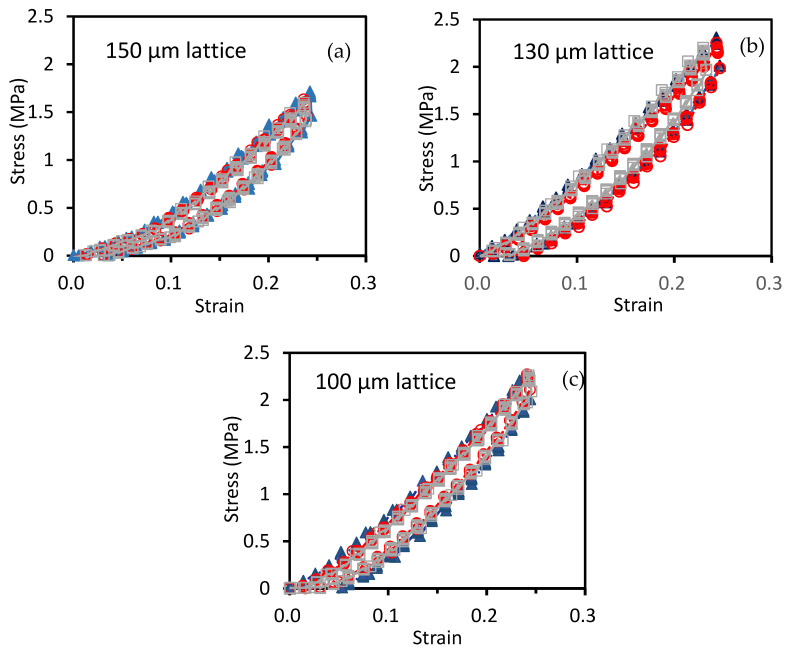
Loading-unloading traces of the micro-lattice structures with unit cell length L of (**a**) 150 μm; (**b**) 130 μm; (**c**) 100 μm.

**Figure 8 polymers-13-00785-f008:**
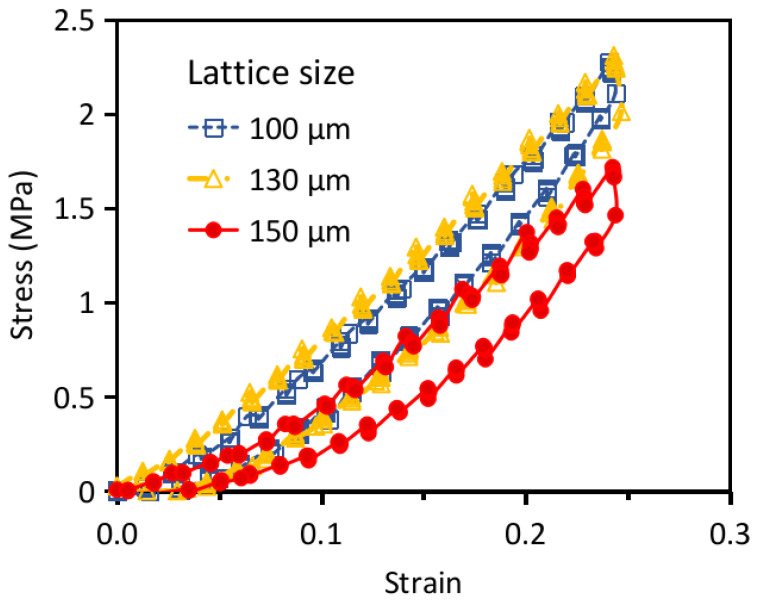
Comparison of the loading-unloading traces of the micro-lattice structures with different unit cell lengths.

**Figure 9 polymers-13-00785-f009:**
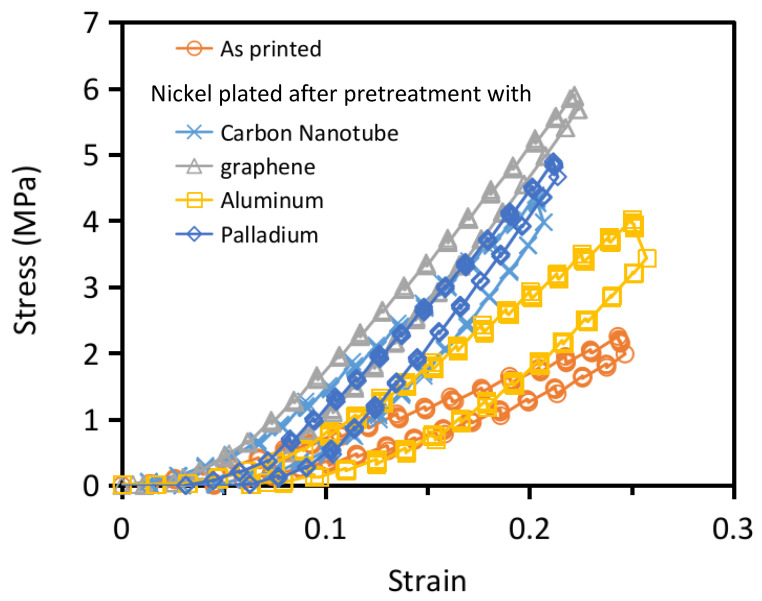
Comparison of the loading-unloading traces of the 130 μm micro-lattice structures with different plating treatments.

**Figure 10 polymers-13-00785-f010:**
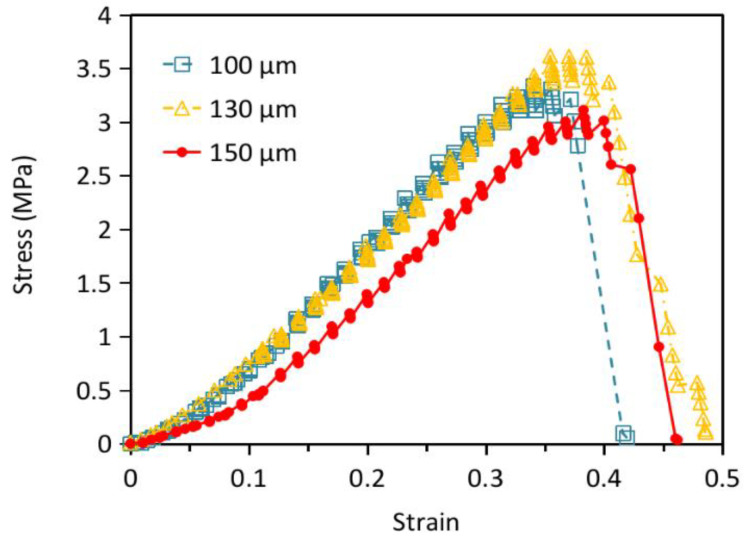
Compressive load-displacement relations to failure of the micro-lattice structures with different unit cell lengths.

**Figure 11 polymers-13-00785-f011:**
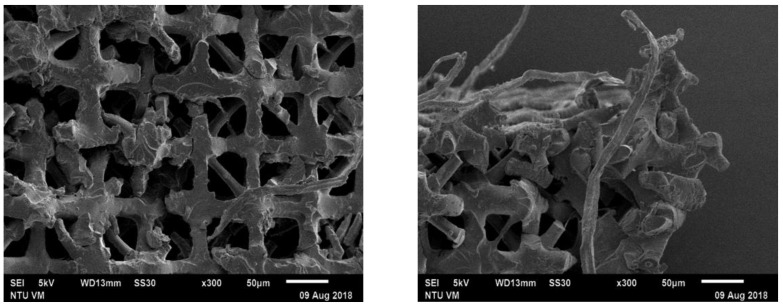
Fracture surface appearance of the 130 μm micro-lattice structure after compressive failure.

**Figure 12 polymers-13-00785-f012:**
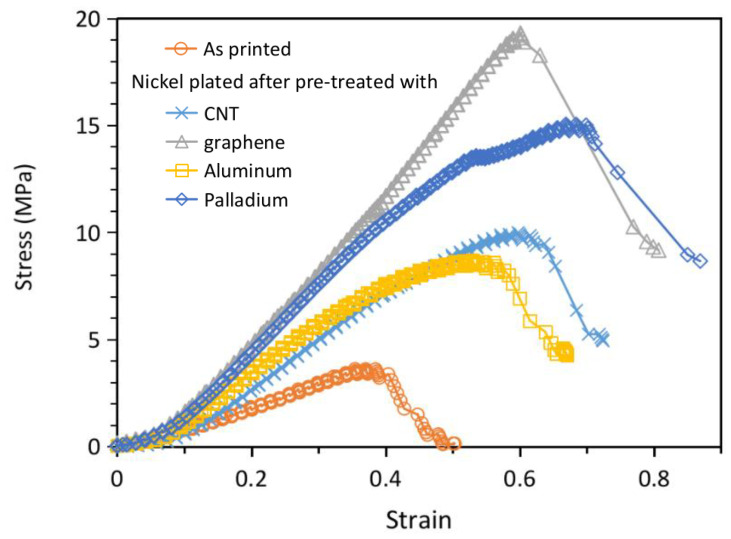
Compressive load-displacement relations to failure of the 130-μm micro-lattice structures with different plating treatments.

**Table 1 polymers-13-00785-t001:** The design and actual dimensions of truss elements for the 3 unit cell sizes printed.

	150 µm Unit Cell		130 µm Unit Cell		100 µm Unit Cell	
	Design	Measured	Design	Measured	Design	Measured
L (µm)	150	160.8 µm	130 µm	135.4 µm	100 µm	105.3 µm
A1 (µm)	15	18.5 µm	13 µm	17.9 µm	10 µm	12.5 µm
A2 (µm)	15	17.0 µm	13 µm	17.0 µm	8 µm	10.3 µm
A3 (µm)	10	13.1 µm	8 µm	11.5 µm	6 µm	9.5 µm

**Table 2 polymers-13-00785-t002:** Dimensional changes after electroless nickel plating using different pretreatments.

		Pretreatment			
	Before Plating	Carbon Nanotube Dip Coating	Graphene Dip Coating	Aluminum Sputtering	Palladium Chloride Activating
		Plating Time: 1 h	Plating Time: 1 h	Plating Time: 0.5 h	Plating Time: 3.5 h
A1 (µm)	17.9	21.29	23.07	23.6	21.2
A2 (µm)	17.0	20.3	22.2	22.8	20.4
A3 (µm)	11.5	14.6	19.8	19.8	15.8

**Table 3 polymers-13-00785-t003:** Atomic % of element detected from energy dispersive spectroscopy analysis.

Elements	Pretreatment							
	Carbon Nanotube Dip Coating		Graphene Dip Coating		Aluminum Sputtering		Palladium Chloride Activating	
	Top	Interior	Top	Interior	Top	Interior	Top	Interior
Carbon	57.08	71.79	33.42	72.54	34.21	70.68	43.32	72.63
Oxygen	21.36	26.21	3.76	23.15	13.12	28.77	5.17	23.73
Phosphorus	4.07	0.44	10.13	0.89	9.83	-	11.12	0.70
Nickel	17.48	1.56	52.70	3.42	42.84	0.56	40.38	2.93

**Table 4 polymers-13-00785-t004:** Compressive stiffness of lattice structures with different unit cell sizes.

150 µm Unit Cell	130 µm Unit Cell	100 µm Unit Cell
8.8 MPa	9.5 MPa	11.1 MPa

**Table 5 polymers-13-00785-t005:** Compressive stiffness after electroless nickel plating using different pretreatments.

As Printed	Pretreatment			
	Carbon Nanotube Dip Coating	Graphene Dip Coating	Aluminum Sputtering	Palladium Chloride Activating
9.5 MPa	27.8 MPa	34.2 MPa	21.6 MPa	32.6 MPa

**Table 6 polymers-13-00785-t006:** Compressive failure load and displacement of the micro-lattice structures with unit cell length of 130 μm with different nickel-plating treatments.

		Pretreatment			
	No Plating	Carbon Nanotube Dip Coating	Graphene Dip Coating	Aluminum Sputtering	Palladium Activating
Failure strength (MPa)	3.63	9.91	19.31	8.58	15.05
Failure strain	0.45	0.63	0.64	0.57	0.69

## Data Availability

Not applicable.

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
