# Peer review of "Fabrication and Compressive Behavior of a Micro-Lattice Composite by High Resolution DLP Stereolithography"

_polymers, 2021, doi:10.3390/polym13050785_

Round 1

Reviewer 1 Report

Manuscript constitutes a really well-prepared and organized piece of work and it is worth considering for publication. Performed investigations have been designed adequately. Obtained results have been clearly presented and, in general, the whole seems to be interesting and valuable. Some improvements which are recommended are as follows:

    • Authors should explain more widely why exactly such pre-treatments methods have been selected in the research.
    • Fig. 7.: Stress-strain characteristics for  structure with unit cell length 150 μm - maximum of y axis should be 2.5 MPa (the same as in the case of the rest of  characteristics) to enable a comparison of the results obtained.
    • Discussion over the results obtained should be to a more extent supported with the comparison with the results of other similar studies (using adequate literature reports). Next, the discussion needs also to be extended by wider explanations of particular observations or statements (e.g. why exactly graphene dip coating pre-treatment resulted in the highest failure strength compared to the other applied pre-treatments? or why structure with 150 μm lattice exhibited the weakest stress-strain characteristic?)
    • Final conclusions should be more specified, e.g. Authors should indicate what was the increase in the stiffness etc.

Author Response

Thank you for pointing out and making the suggestions that helps to improve the readability of the paper.  Corresponding changes have been made in response to each comments as follows.

  • Authors should explain more widely why exactly such pre-treatments methods have been selected in the research.

An explanation of choice of the pre-treatment method is added in lines 158-161. It reads “Method 3 is the conventional pre-treatment for plating on plastic components. It is fairly time consuming and so the more expedient Methods 1 and 2 are attempted with easily available conducting nano materials. Sputtering is known to reliably coat a very thin layer of metal on a component surface and so it is also employed in Method 4 for comparison. ”

  • 7.: Stress-strain characteristics for  structure with unit cell length 150 μm - maximum of y axis should be 2.5 MPa (the same as in the case of the rest of  characteristics) to enable a comparison of the results obtained.

Thanks for point out this disparity. The y axis for the structure with unit cell length 150 μm has been harmonized with the other two graphs as suggested.

  • Discussion over the results obtained should be to a more extent supported with the comparison with the results of other similar studies (using adequate literature reports). Next, the discussion needs also to be extended by wider explanations of particular observations or statements (e.g. why exactly graphene dip coating pre-treatment resulted in the highest failure strength compared to the other applied pre-treatments? or why structure with 150 μm lattice exhibited the weakest stress-strain characteristic?)

To enhance the discussion, the stiffnesses of different structures has been computed and listed in new Tables 4 and 5. Corresponding descriptions have been added to the text: “Table 4 lists the stiffnesses computed after a strain of 10% of these structures. The stiffness steadily inceases from 8.8 MPa to 11.1 MPa as the unit cell size decreases from 150μm to 100μm. ” (lines 284-286) and “Table 5 compares the stiffnesses computed after a strain of 10% of these structures. The graphene treated structure exhibited the largest stiffness of 34.2 MPa, which is 3.6 times that of the un-plated structure.” (lines 297-299).

To explain the observation that the 150 μm lattice exhibited the weakest stress-strain characteristic and stiffness increases with decreasing cell size,  “This may be attributed to the increased number of unit cells and truss elements to bear the load within the same volume as the cell size is reduced.” is added.(lines 286-288). The better stiffness and strength of the graphene treated structure may be attributed to better coverage of nickel plating following this pre-treatment and this explanation is offered in Lines 299-303 and Lines 341-343.  

For comparison with other similar studies, the following has been added: “Song et al. have also fabricated stereolithographic lattice material composite metallized with nickel using the palladium chloride pre-treatment [48]. Direct comparison of the stiffnesses and strengths with the current results may not be meaningful as the resin materials and the lattice designs are different. However it is interesting to note that they reported a 1.29 and1.38 times increase in stiffness and strength respectively after nickel plating. These increases are considerably smaller than those observe in the current case. A possible reason is their large lattice unit cell (~7.5mm in length) and struts (diameter ~1mm) limited the surface area and led to a  much smaller nickel to resin ratio, so nickel contributed much less to increase the stiffness and strength.” (Lines 348-357)

  • Final conclusions should be more specified, e.g. Authors should indicate what was the increase in the stiffness etc.

To make the conclusions more specific, the following sentences have been added: “ By reducing the cell size from 150μm to 100μm increased the compressive stiffness from 8.8MPa to 11.1MPa. ”  (Lines 372-373) “Although the amount of nickel plated into the interior of the structure is sparse and incomplete, failure strength and stiffness has been increased by 5.3 and 3.6 times respectively in the best case of graphene pre-treatment, and 2.4 and 2.3 times respectively in the worst case of aluminum sputtering.” (Lines 377-379)

Reviewer 2 Report

Dear Editor

In the current study, the authors have reported the development of a low-cost micro-stereolithographic system based on a commercial DLP projector. The Scientific soundness of the study is up to the standard. The experimental design is also fine. The results are well-presented and discussed with references.

I have few minor comments.

-The abstract section is not well managed and poorly presenting the whole study. Please re-write this part (main problem, the proposed solution in the current study, a short view of methodology, and main results).

-Introduction (re-write): Please give a clear hypothesis as a separate paragraph. Avoid extensive discussions as it is making the whole point more confusing. 

-The presentation of SEM images is not very well managed. Please reformat the SEM images. You can make it one or two images. 

-The degree sign, please use the correct degree sign throughout the text.

-Materials and methods: If possible please divide it into sub-sections, so the readers can follow them easily.

-Avoid giving references in the middle of the sentence. cite at the end of the sentence.

-Extend the conclusion section by giving the possible application areas in different fields. 

Author Response

Thank you for pointing out and making the suggestions that helps to improve the readability of the paper.  Corresponding changes have been made in response to each comments as follows.

-The abstract section is not well managed and poorly presenting the whole study. Please re-write this part (main problem, the proposed solution in the current study, a short view of methodology, and main results).

The abstract has been modified to state the problem encountered, our current approach and methodology as well as the major results as: “Lattice structures are superior to stochastic foams in mechanical properties and are finding increasing applications. Their properties can be tailored in a wide range through adjusting the design and dimensions of the unit cell, changing the constituent materials as well as forming into hierarchical structures. In order to achieve more levels of hierarchy, the dimensions of the fundamental lattice has to be small enough. Although lattice size of several microns can be fabricated using the two-photon polymerization technique, sophisticated and costly equipment is required. To balance the cost and performance, a low-cost high resolution micro-stereolithographic system has been developed in this work based on a commercial Digital Light Processing (DLP) projector. Unit cell lengths from 100-150μm have been successfully fabricated. Decreasing the unit cell size from 150μm to 100μm increased the compressive stiffness by 26%. Different pre-treatments to faciliate the electroless plating of nickel on the lattice structure has been attempted. A pre-treatment of dip coating in a graphene suspension is the most successful and increased the strength and stiffness by 5.3 and 3.6 times respectively. Even a very light and incomplete nickel plating in the interior has increase the structural stiffness and strength by more than twofold”

-Introduction (re-write): Please give a clear hypothesis as a separate paragraph. Avoid extensive discussions as it is making the whole point more confusing.

An hypotheis has been added and the last paragraph of introduction has been modified to: “With smaller lattice cell size, more cells and structural elements can be packed in the same volume. Moreover, smaller cell size leads to a higher surface to volume ratio and more effective surface metallization. This should result in more marked strengthening effect. To test the above hypothesis, a low-cost high resolution micro-stereolithographic system based on a commercial digital light processing (DLP) projector will be developed. Micro-lattice with cell size down to a hundred microns will be attempted to investigate the effect on stiffness and failure strength of cell size as well as surface metallization with electroless nickel plating” (Lines 91-98)

-The presentation of SEM images is not very well managed. Please reformat the SEM images. You can make it one or two images.

Sorry we are not clear with the exact meaning of this suggestion. We would like to receive further advice before making changes.

Does it mean there are too many SEM images? Currently there are four figures involving SEM images and for completeness’s sake, each of them are key to show the specimen situation in the corresponding stage.

If this suggestion is about the crowding together of several images. We can rearrange them in a single column. However, our original intention to put the images side by side as close as possible is to facilitate comparison the effect of different parameters such as size and pre-treatment.

-The degree sign, please use the correct degree sign throughout the text.

The degree sign has been changed to the degree C sign font instead of combining a superscripted o and C. (Line 166)

-Materials and methods: If possible please divide it into sub-sections, so the readers can follow them easily.

Materials and methods has ben subdivided into 5 sub-sections, namely: 2.1. Micro-stereolithographic printing system; 2.2. Lattice design; 2.3. Photo-curable Resin; 2.4. Nickel plating of the micro-lattice structure; 2.5. Compression testing of the micro-lattice structure.

-Avoid giving references in the middle of the sentence. cite at the end of the sentence.

References in lines 52, 121, 191 have been moved to the end of the sentence. Some references that follows the enumeration a number of different entities have been left next to the individual entity so as to direct interested readers to the correct literature sources.

-Extend the conclusion section by giving the possible application areas in different fields.

A paragraph summarizing the key contributions of the current work and their implication on the application of lattice structures has been added as follows: “The results of the current work shows that it can be fabricated in a low cost way to high resolution and that its mechanical properties is not limited by the cured resin but can be easily improved by a large extent by surface metallization. The small cell size also facilitated the fabrication of hierarchical lattice structures, which is a further way to modulate the mechanical properties. With the availability of wider property range, their applications in impact energy absorption, vibration damping, biomedical implant and structural designs that require light weight can become more flexible.”